# CodeCMR: Cross-Modal Retrieval For Function-Level Binary Source Code Matching

**Zeping Yu**[1], **Wenxin Zheng**[12], **Jiaqi Wang**[1], **Qiyi Tang**[1], **Sen Nie**[1], **Shi Wu**[1]*
[1]Tencent Security Keen Lab, Shanghai, China
[2]Shanghai Jiao Tong University, Shanghai, China
{ravinyu, dodgetang, snie, shiwu}@tencent.com {zwx9810, yakkiwa}@gmail.com

## Abstract

Binary source code matching, especially on function-level, has a critical role in the field of computer security. Given binary code only, finding the corresponding source code improves the accuracy and efficiency in reverse engineering. Given source code only, related binary code retrieval contributes to known vulnerabilities confirmation. However, due to the vast difference between source and binary code, few studies have investigated binary source code matching. Previously published studies focus on code literals extraction such as strings and integers, then utilize traditional matching algorithms such as the Hungarian algorithm for code matching. Nevertheless, these methods have limitations on function-level, because they ignore the potential semantic features of code and a lot of code lacks sufficient code literals. Also, these methods indicate a need for expert experience for useful feature identification and feature engineering, which is time-consuming. This paper proposes an end-to-end cross-modal retrieval network for binary source code matching, which achieves higher accuracy and requires less expert experience. We adopt Deep Pyramid Convolutional Neural Network (DPCNN) for source code feature extraction and Graph Neural Network (GNN) for binary code feature extraction. We also exploit neural network-based models to capture code literals, including strings and integers. Furthermore, we implement "norm weighted sampling" for negative sampling. We evaluate our model on two datasets, where it outperforms other methods significantly.

## 1 Introduction

Binary source code matching is an essential task in computer security. Most applications focus on binary to source matching, including code clone detection [1], open-source code reuse identification [2, 3, 4], and reverse engineering [5, 6]. At the same time, source to binary matching is also useful. When we have source code, it is significant for us to know whether its corresponding binary code is included in a binary file, so that we could use it for vulnerability risk warning. The binary source code matching task is difficult because the differences between source code and binary code are enormous. Previous methods usually extract the code literals, including strings, integers, if/else numbers [2], recursions [5], etc. Then traditional matching algorithms, such as the Hungarian algorithm [7], are adopted to compute the code similarity. However, these methods have two main problems. Firstly, they could not achieve high accuracy, because only the code literals of the code are used. The potential features, which may contain much more information, are ignored. Secondly, these methods need expert experience to choose the features and do feature engineering, which costs a lot of time.

Our model is built on function-level. Compared with previous library-level tasks [1, 2, 3, 4, 5, 6], function-level inputs have fewer strings, and fewer integers, so the designed model needs to be more

---

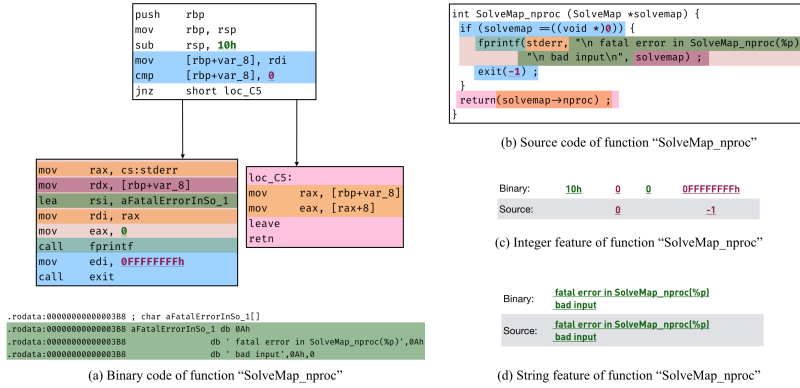

Figure 1: An example of a source-binary code pair. (a) shows the binary code, and (b) shows its corresponding source code. (c) and (d) are the literals of the source and binary code, which are integers and strings.

precise. For ease of understanding, we show a function-level source-binary code pair in Figure 1. The strings are the same in the source code and the binary code, but this feature is coarse-grained because the strings are few and different functions may have the same strings. The set of integers is another useful feature, but they are not equal because address numbers may appear in binary code. Also, it is a coarse-grained feature because the same integers may appear in different functions. As a consequence, more features should be extracted to increase the accuracy of code matching. From Figure 1 we could see that there are many potential semantic features between the source code and the binary code. For example, the function call in the source code could be matched with the lower left block in the binary code's CFG. To show more clearly, we color the matching code in both source and binary code. This is similar to cross-modal tasks, such as image caption. Even if the data belongs to different formats, the potential semantic relationship still exists. If these semantic features could be extracted, the matching accuracy will be significantly improved.

Based on these insights, we propose an overall framework for binary source code matching. We first put this scenario into the cross-modal retrieval task, whose modalities are source code and binary code. Cross-modal retrieval is used in many applications, including text-image [8, 9], audio-image [10], text-audio [11, 12], etc. With the development of deep learning in recent years, a general framework for cross-modal retrieval tends to be popular. The two modalities are first put into modality-specific encoders and computed into two vectors, and then different negative sampling methods and different loss functions are designed in respective tasks. For example, in the recipe-image task [8], they adopt Hierarchical-LSTM [13] and Resnet [14] to extract the recipe feature and the image feature separately, then use adversarial loss, triplet loss and hard sample mining for better training.

After proposing the overall framework, we design encoders for extracting the features of source code and binary code. For source code, we use the deep pyramid convolutional neural network (DPCNN) [15]. We find DPCNN powerful for character-level source code sequences. For binary code, we design a graph neural network (GNN) method. Instead of using pre-training methods for node embeddings [16], we build an end-to-end HBMP model [17] to generate the node embeddings, and then use GGNN [32] and Set2Set [33] model to compute the semantic embedding of the binary code. Also, we exploit two models for strings and integers to extract the code literal features. We build an LSTM model taking integers as input to capture the volatile integers, and produce a hierarchical-LSTM model for strings. We combine these three features for both source code and binary code. Additionally, we design a norm weighted sampling method, which could accelerate training and significantly increase accuracy.

Our contributions are as follows:

1) We define the function-level binary source code matching problem, which is vital in the computer security area. We categorize this problem into the cross-modal retrieval task and propose an overall framework to solve it.

2) For source code, we use the DPCNN model on character-level, which leads to a satisfactory performance. Also, using character source code as input is more robust and faster than parsing the code into an abstract syntax tree (AST). For binary code, we produce an end-to-end model instead of pre-training methods, which increases both accuracy and efficiency. The models can extract the potential semantic features of source and binary code, thereby significantly improving the accuracy.

3) We propose methods for extracting code literals, including strings and integers. Meanwhile, we design a new negative sampling method called "norm weighted sampling", which accelerates training and improves results.

4) We conduct experiments on two datasets. On both datasets our model outperforms other models a lot. In particular, on our dataset with 10,000 samples, the recall@1/recall@10 result achieves 90.2%/98.3%, which meets the requirement of actual use.

## 2 Related work

### 2.1 Binary source code matching

Binary source code matching is an essential task in computer security. Most work focuses on binary to source matching, and tries to extract the literals of source and binary code. BinPro [5] extracts the matching features and predictive features, and uses the Hungarian algorithm to match the code. B2SFinder [2] utilizes seven traceable features and employs a weighted feature matching algorithm. RESource [3] draws inspiration from Re-Google, which uses the string literals and trigger queries based on certain repositories used by the developers' community. The Binary Analysis Tool (BAT) [1] applies unpack scans and leaf scans to extract information, and proposes a data compression method to compute the similarity score. OSSPolice [4] exploits string literals and exported functions as features, and employs software similarity comparison to detect OSS reuse. Codebin [6] defines a method to compute the complexity of control flow graphs (CFGs), and extracts features from the function calls and CFGs.

Different from these library-level binary source code matching methods, our research is on function-level. Since a function is much smaller than a library, the literals on binary code and source code are much fewer. For example, the number of strings and integers drops a lot; function call features and symbol tables cannot be used. Consequently, the extracting features need to be more fine-grained and the model needs to be more precise.

### 2.2 Code representation learning

Apart from binary source code matching, research for source code representation and binary code representation is also important. For source code, SourcererCC [18] takes the token sequences as input features. ASTNN [19] splits each large AST into many small statement trees, encodes the statement trees to vectors, and uses a bidirectional RNN model to compute the representation. For binary code, traditional methods utilize graph matching algorithms [20]. A neural network-based approach called Gemini [21] proposes a GNN model to learn the binary embedding. Yu et al. (2020) uses BERT pre-training methods [42] to learn the semantic features from the node embeddings. Zuo et al. (2018) proposes a neural machine translation method to learn the semantic relationship.

### 2.3 Cross-modal retrieval

Cross-modal retrieval aims to retrieve the relevant instances from different modalities [8]. One general method is to compress the cross-modal inputs into different embeddings by modality-specific encoders, and then design loss functions to learn the similarities and reduce loss. Generating vectors in common space is the main challenge[23], and many efforts are made to solve it. Several research uses metric learning methods to learn the similarities [24, 25], and some uses adversarial training methods [26] to correlate the cross-modal embeddings [8, 9, 27, 28]. CCA [29] proposes the canonical correlation analysis method to learn the similarity between text and image. DCCA [30] uses deep networks to learn the nonlinear transformation between the two modalities so that the results are highly linearly related. Besides, the alignment methods are also commonly used, including local alignment [31] and global alignment [30].

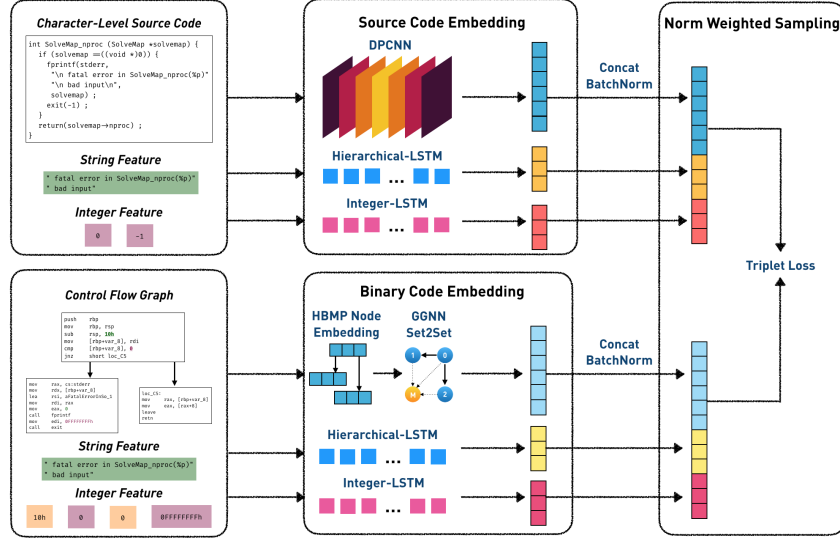

Figure 2: The overall framework of CodeCMR. The source code and binary code are encoded by DPCNN and GNN to compute their semantic features, and two neural network-based models are also used to extract strings and integers. Then the norm weighted sampling method is designed to get negative samples, and triplet loss is chosen to be the loss function.

# 3 Proposed methods

## 3.1 Overall framework

The overall framework of our proposed methods is shown in Figure 2. Both source code and binary code have three inputs: semantic input (character-level source code and CFGs), string input, and integer input. These inputs are fed into different encoders. For character-level source code, we use DPCNN [15] with the global average pooling method to extract the semantic features. For CFGs, we propose a GNN model with HBMP node embeddings, using GGNN [32] message passing method and Set2Set [33] graph pooling method to compute the graph embedding. Also, we propose two neural network-based models to extract the string features and integer features. After calculating the embeddings of the three inputs of source code and binary code, we adopt a simple alignment method, that is, concatenate and batch normalization [34]. Then we design a new negative sampling method called "norm weighted sampling", which could change the selection probabilities based on each distribution. At last, we use the triplet loss [35] to make the positive pair's similarity larger than the negative pair's similarity. We want to assure that the distance D(A, P) between the anchor (source/binary code) and the positive sample (its corresponding binary/source code) is closer than the distance D(A, N) between the anchor and the negative sample (any other binary/source code). And a distance margin is used to separate the positive pair from the negative.

$$Loss = Max(D(A,P) - D(A,N) + margin, 0) \qquad (1)$$

More generally, many improvements could be added to this framework for extending research, whether using the features as pre-trained weights or adding a new task for multitask models. For example, the proposed code features could be used for classification tasks such as code clone detection, and the code representations may be used as the initial embeddings for generative models. Also, adding adversarial loss on alignment may lead to a higher similarity between the source and binary code embeddings. For generative tasks such as generating source code based on binary code, adding adversarial loss may be useful.

## 3.2 Code semantic features

The models for extracting the code semantic representation are shown in Figure 3. Figure 3 (left) shows the structure of the DPCNN model for source code. The source code is first preprocessed

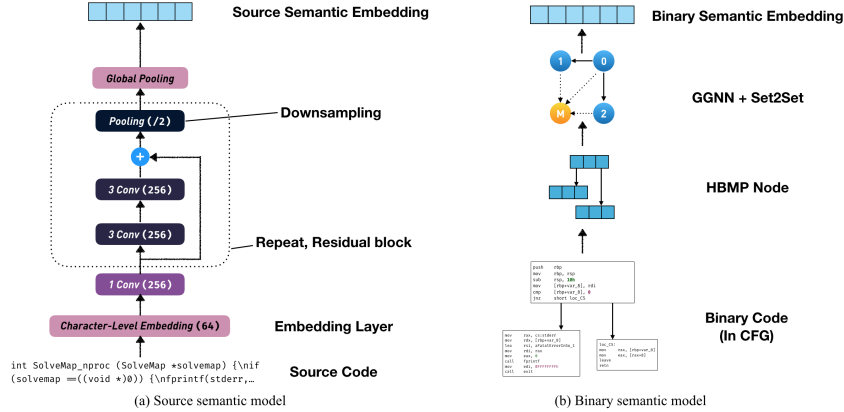

Figure 3: Models for extracting code semantic representation on source and binary code. For source code, DPCNN is used on character-level source code sequences. For binary code, a proposed GNN model with HBMP node embeddings and GGNN + Set2Set model is used on CFGs.

into a long character-level sequence and fed into an embedding layer to compute the character embeddings. The weights of the embedding layer are learned during training. In our implementation, we replace the region embedding [36] with a convolutional layer with kernel size 1 and match the dimensions between the embedding layer and the residual layers. Inspired by ALBERT [37], we set the embedding layer's dimension smaller than the residual layers' to save parameters. Then the embeddings are taken as the input of several repeating DPCNN blocks. Each block has two residual convolutional layers and one downsampling pooling layer (except the last block). The convolutional layers share the same feature dimensions, which is beneficial for both storage and time efficiency [15]. At last, we use a global average pooling layer to generate the final representation.

For binary code, the CFG is taken as input. Instead of using pre-trained block embeddings [16], we build an end-to-end model to train the node embeddings together. At the first step the node tokens are transformed into token embeddings. The block representation is computed by the HBMP model [17], which has three LSTM layers, and each layer takes the output of the previous layer as the initial hidden state. After obtaining the block representation, we use GGNN [32] to pass the messages of the adjacent nodes by several iterations. At last, we use Set2Set [33] to compute the final CFG representation. Note that the Set2Set graph pooling method is order-independent, which means the result would be invariant even if the node order changes. This is also accordant with the order-independent global average pooling layer of DPCNN on character-level source code.

In our paper, we use DPCNN for source code and HBMP + GGNN + Set2Set for binary code to extract the semantic representation. For real use, other methods such as ASTNN [19] could also work. Here, we choose DPCNN because of no required extra work. If using ASTNN, source code should be parsed into ASTs, which will cost much time. In some cases, the parsing method cannot extract all the code. For example, when we use the gcc parsing method [19] on our dataset, only 10% of the code are successfully parsed. Even though we change a parsing method and solve the problem, the requirement of more expert experience and time cannot be ignored.

### 3.3 Code literal features

Apart from the semantic features, the literal features could also play a role. As shown in Figure 1, the integers are similar in source code and binary code, but they are not identical due to the existence of extra address numbers in binary code. Another situation should also be taken into consideration, which is, the integers may be altered. For example, "i<=5" may be changed into "i<6", and "x*16" may be changed into "x«4". Also, some integers will expand or shrink by two times on different platforms. So when we design our model, we should consider not only the integer tokens but also the integer numbers.

Based on this idea, we propose an LSTM-based model named integer-LSTM. We first sort the sequences and put the integer tokens into an embedding layer. The integer-LSTM takes two features

as input: the integer token and the integer number. The integer number is utilized in both input gate and output gate to control the amount of used information. The equations are as follows:

$$f_k = \sigma(x_k W_f + h_{k-1} U_f + b_f) \tag{2}$$

$$i_k = \sigma(x_k W_i + h_{k-1} U_i + log(x_{kn}) \cdot N_i + b_i) \tag{3}$$

$$o_k = \sigma(x_k W_o + h_{k-1} U_o + log(x_{kn}) \cdot N_o + b_o) \tag{4}$$

$$c_k = f_k \odot c_{k-1} + i_k \odot \phi(x_k W_c + h_{k-1} U_c + b_c) \tag{5}$$

$$h_k = o_k \odot \phi(c_k) \tag{6}$$

where $x_k$ is the integer token embedding, $h_k$ is the hidden state, and $x_{kn}$ is the integer number. The integer number $x_{kn}$ works at the input gate and the output gate. If the current integer is regarded useless for prediction, the model could set the input gate or output gate to zero. After getting the hidden states of the integer-LSTM, we use an average pooling layer on all the hidden states to extract the information, so that the output gate could also make contribution. In practice, we concatenate the token embedding and the log number score as the new input embedding, in order to provide the model more information.

For strings, we build a simple hierarchical-LSTM model. Since the set of the strings are the same in source and binary code, any model could produce the same embedding for source and binary code. However, our model is more powerful because it could allocate larger similarity scores to similar strings. We first use an LSTM model on each string to compute the string embedding and then use a sum pooling method on top of the strings to calculate the embedding of the string set. Take Figure 1 as an example, the string set of source code and binary code is the same, which contains two strings: "fatal error in SolveMap_nproc(%p)" and "bad input". The model first use the LSTM model to convert the two strings into two embeddings, and then use a sum pooling layer on the two string embeddings to compute the set embedding.

## 3.4 Norm weighted sampling

After computing the three feature embeddings of source and binary code, we concatenate the features and add a batch normalization layer to get the alignment embeddings. To make the true pair's similarity larger than the false pair's similarity, we use the triplet loss. However, triplet loss has the negative sampling problem, which means different negative samples may cause very different results [35]. Schroff et al. (2015) indicates that the hard examples would lead to bad local minima in early training, so they adopt a semi-hard negative sampling method. However, despite the quicker convergence at the beginning, semi-hard sampling may stop progressing at some points [38]. Wu et al. (2017) proposes a distance weighted sampling method. The sampling method is as follows:

$$w_{logk} = -(n-2)d_k - \frac{n-3}{2}(1 - \frac{1}{4}d_k^2) \tag{7}$$

$$[w_1, w_2, ..., w_i] = softmax([w_{log1}, w_{log2}, ..., w_{logi}]) \tag{8}$$

where n is the dimension number, and $d_k$ is the kth sample's distance. By computing equation (7) and (8), the samples are weighted by the distances. The chosen probability of the kth sample is equal to $w_k$. The distance weighted sampling method could select from a wide range of samples, while other techniques including uniform sampling, hard sampling, and semi-hard sampling, may only select within certain distance [38].

The norm weighted sampling method makes an improvement in distance weighted sampling. It adds a hyperparameter $s$ to control the probabilities expanding or shrinking.

$$[w_1, w_2, ..., w_i] = softmax([w_{log1}s, w_{log2}s, ..., w_{logi}s]) \tag{9}$$

The idea is similar to NormFace [39]. In face recognition, adding the scaling parameter $s$ after normalization is a common method [39, 40], because it could scale the softmax scores and accelerate the convergence. In our task, adding the parameter $s$ could help expand or shrink the probability scores. For example, a number sequences computed by equation (7) is [-0.74, -1.00, -0.95, -0.40, 0.00]. When $s = 1$ (the same as distance weighted sampling), the selecting probabilities are [0.16, 0.12, 0.13, 0.23, 0.36]; when $s = 2$, they are [0.12, 0.07, 0.08, 0.23, 0.50]; when $s = 0.5$, they are [0.18,

0.16, 0.17, 0.22, 0.27]. We could observe that with the ranking order of the probabilities unchanged, the score of the largest probability enlarges or shrinks, controlled by $s$. In practice, we could change the hyperparameter $s$ to adapt to different tasks and different datasets. The hyperparameter $s$ could also be seen as a negative temperature parameter, sometimes denoted beta.

# 4 Experiments

## 4.1 Datasets and implement details

We evaluate our model on the binary source code matching task. The source code could be compiled into CFGs with different compilers, different platforms and various optimizations. We choose gcc-x64-O0 and clang-arm-O3 as two combination methods to build two datasets. Each dataset contains 30,000 source-binary pairs for training, 10,000 pairs for validation, and 10,000 pairs for testing. For binary code, the IDA Pro tool is used to extract the tokens and features. We use recall@1 and recall@10 as two evaluation metrics, which means the similarity score of the true pair ranks as top1/top10 in all the 10,000 pairs' similarities on testset. More information about the dataset could be found at https://github.com/binaryai.

For the training process, the training epoch is set to 64 for gcc-x64-O0 and 128 for clang-arm-O3. The learning rate is 0.001, the batch size is 32, the triplet margin is 0.5, and the optimizer is Adam. For source code, the length of character-level sequences is 4,096; the dimension is 64 on embedding layer and 128 on convolutional layers. The repeat number of residual blocks is 7. For binary code, the dimension of node embedding and graph embedding is both 128; the number of GGNN message passing iteration and Set2Set iteration are both 5. For strings and integers, the embedding layers' dimension and LSTM's hidden dimension are both 64.

## 4.2 Compared methods

**Traditional methods** We choose BinPro [5] and B2SFinder [2] in comparison with our model, because these two methods extract more features. We use the Hungarian algorithm [7] on strings and integers to compute the matching scores.

**Source code semantic feature** Our model uses the DPCNN model with global average pooling to extract the semantic features from character-level source code. Besides, we also choose LSTM and TextCNN [41], which are two common methods in NLP.

**Binary code semantic feature** Our binary model is an end-to-end model, in which the block embeddings are generated during training, while other methods use manually-designed features [21] or pre-trained embeddings [16]. Since Yu et al. (2020) has shown that pre-training methods could achieve better results than manually-designed features, we choose BERT [42] and Word2vec [43] as the comparison.

**Code literal feature** We compare our integer-LSTM model with the traditional LSTM model to explore the ability of adding the integer number. Also, we compare the neural network-based methods with the Hungarian algorithm.

**Negative sampling methods** Since Wu et al. (2017) has proved that distance weighted sampling could get better results than semi-hard sampling, we compare our model with distance weighted sampling and random sampling. We explore the role of $s$ when taking different values.

## 4.3 Results

Table 1 shows the results of all the models. The first section of Table 1 contains traditional methods. The second section ignores the code literals and the third section only uses code literals. The last section shows the effect of the norm weighted sampling method, the "Random" line represents the combination of the best models on code semantic feature and code literal feature with random sampling method, which is DPCNN+HBMP+Integer-LSTM(Integer)+Hier-LSTM(String).

Compared with BinPro, B2SFinder, and the Hungarian algorithm, our model has much better results (recall@1 from 38.6% to 90.2%). In comparison with the code literal feature models, the code semantic feature models achieve much higher recall scores. This indicates that the potential semantic features between source and binary code are more important for matching than the code literals.

Table 1: Experiment results on two datasets. "Binary2Source" and "Source2Binary" means using the binary/source code column of embeddings as queries to search the other column. The scores (%) are recall@1/recall@10. Except the last block, the sampling method is random sampling.

| Model | gcc-x64-O0 | | clang-arm-O3 | |
| --- | --- | --- | --- | --- |
| | Binary2Source | Source2Binary | Binary2Source | Source2Binary |
| BinPro | 38.6 / 41.1 | 39.2 / 41.8 | 39.4 / 42.1 | 39.7 / 42.4 |
| B2SFinder | 34.1 / 39.6 | 34.4 / 39.8 | 33.5 / 39.2 | 34.2 / 39.5 |
| TextCNN + HBMP | 54.3 / 84.7 | 54.7 / 85.1 | 48.8 / 82.5 | 49.3 / 82.8 |
| LSTM + HBMP | 63.7 / 89.4 | 63.9 / 88.7 | 60.2 / 86.9 | 60.6 / 87.3 |
| DPCNN + Word2vec | 69.2 / 91.0 | 69.6 / 90.7 | 63.6 / 88.3 | 64.0 / 88.5 |
| DPCNN + BERT | 74.3 / 93.9 | 74.5 / 94.0 | 66.1 / 89.0 | 66.5 / 89.5 |
| DPCNN + HBMP | 80.8 / 96.4 | 81.2 / 96.6 | 72.9 / 91.2 | 73.2 / 92.1 |
| Hungarian (Integer) | 9.0 / 15.6 | 11.6 / 17.7 | 7.7 / 14.3 | 10.4 / 17.3 |
| LSTM (Integer) | 10.7 / 17.9 | 13.0 / 19.0 | 8.9 / 15.2 | 10.9 / 18.6 |
| Integer-LSTM (Integer) | 12.3 / 19.4 | 15.5 / 23.2 | 11.5 /17.1 | 12.2 / 20.7 |
| Hungarian (String) | 33.9 / 35.4 | 34.0 / 35.5 | 35.6 / 36.2 | 35.8 / 37.3 |
| Hier-LSTM (String) | 42.4 / 44.5 | 42.8 / 45.1 | 45.0 / 46.9 | 45.5 / 48.7 |
| Random | 81.9 / 97.3 | 82.3 / 98.0 | 74.2 / 92.0 | 74.8 / 92.6 |
| Distance-Weight | 86.2 / 97.4 | 86.5 / 97.8 | 77.4 / 94.3 | 78.2 / 94.7 |
| Norm-Weight ($s = 0.5$) | 85.3 / 97.2 | 85.4 / 97.5 | 76.9 / 93.4 | 77.5 / 93.6 |
| Norm-Weight ($s = 2$) | 89.0 / 97.9 | 89.1 / 98.2 | 81.2 / 95.1 | 82.5 / 95.4 |
| **Norm-Weight** ($s = 5$) | **90.2 / 98.3** | **90.3 / 98.5** | **87.3 / 97.5** | **87.7 / 97.9** |

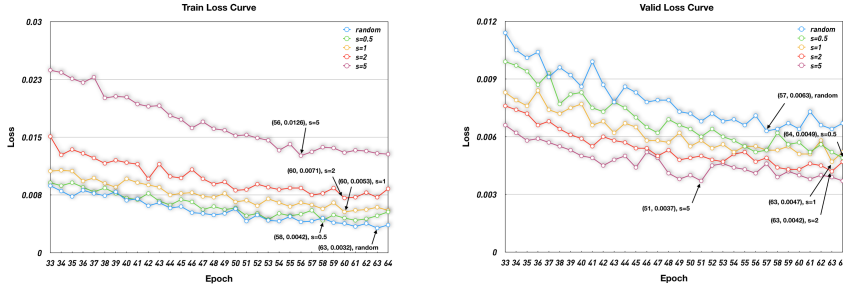

Figure 4: Train/valid curves of different negative sampling methods on gcc-x64-O0 dataset.

Focusing on code semantic models, DPCNN + HBMP achieves the best performance. This demonstrates that a deep end-to-end model has more capacity to extract information than a shallow pretrained model (DPCNN vs. TextCNN/LSTM, HBMP vs. BERT/Word2Vec). For code literals, integer-LSTM has better performance than traditional LSTM because of the integer number feature. And deep learning models have more capacity than the traditional methods for extracting the literal features of code.

To learn the effect of norm weighted sampling, we set $s$ to 0.5, 2, and 5. The highest recall@1 score (when $s = 5$) has about 8% improvement than random sampling. We also observe that larger $s$ contributes to better result in our task. For further exploration, we draw the train/valid curves to reveal the changes in loss. Note that because of the triplet loss, if the anchor-negative distance is larger than anchor-positive distance with margin, the triplet pair has no contribution to training. But it also participates in forward propagation, and also be calculated when computing the loss. Therefore, in the train/valid curves, although the random sampling method has smaller train loss due to the useless triplet selection, its valid loss is larger. In contrast, our norm weighted sampling method has larger train loss because of more useful triplet selection, especially when $s$ is larger. However, its valid loss is smaller and recall score is higher. We also conduct experiments on larger s (s=8 and s=10), and we notice that the score declines slightly, but not too much. So this means s=5 is the best choice on this task and this dataset.

# 5 Discussion

As we have mentioned in section 3.1, many improvements could work based on our proposed framework. In practice, we have tried some novel methods and achieved good results, including Circle loss [44], Cross-batch memory [45], Adversarial loss [26], and Cross-lingual language model [46]. We will discuss more in this section.

**Code encoders** Other encoders could also work on our framework. For source code, ASTNN [19], Tree-LSTM [47] or other tree-based models could be used on ASTs. For binary code, GNN models have achieved good results, but other NLP models such as transformer [48] also worth trying. For code literals, other RNN-based models [49, 50] may work as well. In our work, we use character-level DPCNN on text because of its robustness: no extra expert experience is needed to design the parser for source code.

**Loss function and sampling methods** In the field of metric learning and deep embedding learning, many loss functions and sampling methods have been designed, which may be useful on this task, including AM-softmax [51], Arcface [52], Multi-similarity loss [53], Circle loss [44], and the Cross-batch memory method [45]. In our experiments, Circle loss and Cross-batch memory performs well.

**Adversarial learning and other cross-modal retrieval methods** In cross-modal retrieval, adding adversarial loss is a general idea to correlate the embeddings of different modalities. A WGAN-GP model [54] could be adopted, and then the problem could be solved by a min-max optimization. Besides, other cross-modal retrieval methods [55, 56] may also work on this task. The WGAN-GP model could help increase the scores on our dataset, but we find that the hyperparameters are very sensitive and should be carefully designed.

**Pre-training methods** In the field of NLP, the BERT-based pre-training methods could help the model acquire more knowledge. In our experiments the BERT pre-training method is worse than end-to-end training. This may be caused by only taking the last layer's static output of the BERT model as binary node embedding. Instead, taking the whole BERT model and fine-tuning may be better. Apart from pre-training on binary code or source code only, cross-lingual pre-training is a better way to merge the information. Due to the architecture our framework, each modality does not have the ability to obtain cross-modal information. Cross-lingual language model [46] could solve this problem. Taking the whole pre-training model and fine-tuning or only taking the embedding layer could both help improve accuracy. However, the model complexity and the time cost increase when using these pre-training methods.

# 6 Conclusion

In this paper, we conduct a deep exploration of function-level binary source code matching. Because function-level code has few literals, only using code literals such as strings and integers could not achieve good results. Therefore, we transfer this task to a cross-modal retrieval problem and propose an overall framework for it. For both source and binary code, we extract the potential semantic features and the code literals, merge them into embeddings, and then use triplet loss to learn the relation. Specifically, we adopt DPCNN on character-level source code and use GNN models on binary code. For code literals, we propose the integer-LSTM for integers and hierarchical-LSTM for strings. Moreover, we propose the norm weighted sampling method, which highly improves the performance. We conduct experiments on two datasets to compare with other methods, and the results demonstrate that our proposed model could achieve satisfying results.

# Broader Impact

**Positive outcomes** The security researchers will benefit from this research. Given a large-scale corpus of source-binary pairs, the researchers could use our technology to find the binary code of a vulnerability signature. Also, they could search the source code of a binary signature to help reverse engineering.

**Negative outcomes** For a malicious attacker, our technology may be used to cause a potential negative outcome.

## Funding Disclosure

We have nothing to disclose for this paper.

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
