[Supplementary Material]

# (Supplementary) CodeCMR: Cross-Modal Retrieval For Function-Level Binary Source Code Matching

**Zeping Yu[1], Wenxin Zheng[12], Jiaqi Wang[1], Qiyi Tang[1], Sen Nie[1], Shi Wu[1]**[*]
[1]Tencent Security Keen Lab, Shanghai, China
[2]Shanghai Jiao Tong University, Shanghai, China
{ravinyu, dodgetang, snie, shiwu}@tencent.com {zwx9810, yakkiwa}@gmail.com

## 1 Time comparison

We show the preprocessing, training and testing time of the models in Table 1. We use 16 CPUs and 4 V100 GPUs for all the models, and the training epoch of all the models is 64.

The Hungarian algorithm does not need a training process, but its testing time is larger than neural network-based models. BinPro and B2SFinder also need to use traditional matching algorithms to compute the similarity scores. In practice, neural network-based methods could compute the embeddings for each function and store them in advance, which could save much time.

In comparison with the pre-training models, the end-to-end models get rid of the pre-training time. Although the training and testing time of the pre-training model decreased slightly, the pre-training process is time-consuming.

Compared with random sampling, our norm weighted sampling method requires more time. To sample suitable triplet pairs, we need to compute the embeddings and the probabilities at each epoch. However, the additional time consumption is worth, because the effect has improved a lot.

## 2 Results on 32 combination columns

Previous work indicates that CFG is a sensitive feature, because it changes greatly in different optimizations. However, the results have declined somewhat in our research, but within acceptable limits. To train a cross-platform model, we extract the token sequences based on IDA Pro microcode IR (Intermediate Representation). The columns are combinations of different compilers (gcc/clang), different platforms (x86/x64/arm/arm64), and different optimizations (O0/O1/O2/O3). The average recall@1 score of the 32 combinations is 88.9%. The lowest recall@1 score among the 32 combinations is 85.1% on gcc-arm64-O3, which is acceptable.

To explore the impact of compilers, platforms, and optimizations, we could analyze Figure 1 in-depth. The histogram shows a decreasing curve from O0 to O3 optimizations, while the scores on distinct compilers and platforms are relatively similar. This means our model is not affected by compiler and platform changes but slightly affected by optimization changes.

## 3 Bad case study

To explore the effect of our model, we do bad case study on both binary2source and source2binary. The cases are shown in Figure 2 and Figure 3. The dotted lines show the semantic relationship between the true pair's source code and binary code. Even though the model does not find the correct function in these bad cases, it could find similar functions, so our model is capable of learning the potential semantic relationship between source code and binary code.

---

[*]Corresponding author.

Table 1: Time consumption (minutes) by each model on preprocessing, training and testing on gcc-x64-O0 dataset.

| Model | Pre-process | Train | Test |
|---|---|---|---|
| BinPro | 106.0 | 4.25 | 54.2 |
| B2SFinder | - | - | 64.2 |
| TextCNN + HBMP | - | 160.3 | 2.5 |
| LSTM + HBMP | - | 251.7 | 3.9 |
| DPCNN + Word2vec | 30.4 | 142.5 | 2.2 |
| DPCNN + BERT | 203.2 | 143.0 | 2.2 |
| DPCNN + HBMP | - | 170.6 | 2.7 |
| Hungarian (Integer) | - | - | 60.8 |
| LSTM (Integer) | - | 113.4 | 1.9 |
| Integer-LSTM (Integer) | - | 113.8 | 2.0 |
| Hungarian (String) | - | - | 45.2 |
| Hier-LSTM (String) | - | 108.8 | 1.9 |
| Random | - | 171.3 | 2.7 |
| Distance-Weight | - | 214.7 | 2.7 |
| Norm-Weight (s=0.5) | - | 213.2 | 2.7 |
| Norm-Weight (s=2) | - | 214.9 | 2.7 |
| Norm-Weight (s=5) | - | 214.8 | 2.7 |

Figure 1: Results on 32 datasets. The training binary column of the model is gcc-x64-O0, and it also has good results on other columns. The results (%) are recall@1/recall@10 scores.

In the binary2source case, the model detects the identical character-level sequences such as "if(!PyArg_ParseTuple_SizeT". Although some details are various, the source code of the true example (function "pkcs12_set_preferred_cipher") and the false example (function "error_check_status") is similar, eg. the "if-return" statement. In the source2binary case, the model captures most of the semantic features. The binary code statements of the true example (function "BGl_checkzd2nodezd2sharingza2za2zzast_checkzd2sharingzd2") and the false example (function "BGl_addzd2gotoszd2zzsaw_gotosz00") are also similar, which are colored in Figure 3.

Figure 2: Case study of function "pkcs12_set_preferred_cipher" (binary2source). The similarity of the true/false pair is 0.9611/0.9645.

Figure 3: Case study of function "BGl_checkzd2nodezd2sharingza2za2zzast_checkzd2sharingzd2" (source2binary). The similarity of the true/false pair is 0.9179/0.9474.