[Reviews · NeurIPS 2020]

Review 1

Summary and Contributions: This paper considers the problem of learning to search for compiled binaries that match a particular source code. While others have considered this problem before, this particular proposal considers function-level search, which ha not been considered before. The idea is as follows. The first is the source code itself, the second is the set of strings in the code, and third is the set of integers in the code. For encoding characters, the authors use DPCNN. For context free grammars, a graph neural network is used. And LSTM is used to encode the sequence of integer literals, and a hierarchical LSTM is used to encode strings. To control the choice of negative samples during learning, the authors adapt a distance-weighted sampling method from the literature, and allow the inclusion of a parameter s that adds some entropy to the distribution of probabilities that each of the various database objects are chosen as the negative samples. Experiments show impressive accuracy. It is possible to get nearly 90% recall@1 in all four of the experiments that were run.

Strengths: The number one positive aspect of the paper is the very impressive accuracy numbers that are reported. I think that one could point to the simplicity of the approach as a positive aspect.

Weaknesses: I thought that the motivation for function-level matching was a bit weak. I would have liked the paper to open with a scenario or two where the his useful, or absolutely necessary. Another issue is that the paper is not necessarily technically deep. Though I hesitate to be too tough on the paper for that reason; being able to get really good results with a simple method may be considered a feature of the approach. Actually, it is surprising to me that the authors treat source code as text, and binary code as a context free grammar. In fact, it even more natural to treat source code as a CFG. Binary code is often very simple, whereas source code typically has complicated constructs.I would have liked some more explanation along these lines. I found it problematic that the paper is not self-contained, at least with respect to the loss function and the negative sampling method used. The authors use terms such as the “anchor” with no definition. In equation (1), we have the “margin” which is not defined. I would say that triplet loss is widely known, but it is not canonical (that is, you can’t just assume that people know what you are talking about, unlike a concept such as an LSTM). Just a few sentences explaining some of these ideas would make the paper much more readable. It was a bit difficult to understand what was being shown in some of the rows in Table 1. In the second section of the table, are you ignoring integers and strings? And in the third section of the same, are you ONLY using integers and strings? If so, these seems a little strange, as I would expect that most readers want to see the effect of adding different types of features, in a cumulative fashion. I think that considering them separately is not too useful. Nowhere in the paper nor in the supplementary material is the data set on which this is tested detailed. Where did the codes come from? The information in Section 4.1 is not enough to have even an intuitive feeling for the type of code that is being searched.

Correctness: Yes.

Clarity: Yes, the paper is generally quite well-written (modulo the issues with the paper not being totally self-contained).

Relation to Prior Work: Yes.

Reproducibility: No

Additional Feedback: As mentioned above, I think that a detailed description of the data used is important. *** AFTER AUTHOR RESPONSE *** I'd like to thank the authors for their response. I feel that what it comes down to: the paper is not technically novel, though the application may be. Is that good enough for a NeurIPS paper? It appears as if it is. Congratulations!!


Review 2

Summary and Contributions: Paper Summary: This paper proposes a cross-modal retrieval method for function-level binary source code matching. It considers the binary source code matching problem as a cross-modal retrieval task. Different semantic features are proposed to represent the features of source and binary code. Experiments on two datasets demonstrate the superior performance of the proposed method.

Strengths: ++It seems interesting to consider the function-level binary code matching as cross-modal retrieval problem. ++The paper is well written and easy to follow.

Weaknesses: 1. The paper aims at proposing a cross-modal retrieval method. However, the comparisons with state-of-the-art cross-modal retrieval methods are missing. For example, Deep Supervised Cross-Modal Retrieval CVPR 2019, Deep Cross-Modal Hashing, CVPR 2017 2. The contributions on the cross-modal retrieval part are very weak. It concatenates the features and add a batch normalization layer to get the alignment embeddings. Triplet loss is designed to achieve cross-modal correlation. The technique part is weak compared with many cross-modal retrieval methods in Computer Vision and Multimedia. I cannot identify the innovation of this part. 3. The paper adopts Deep Pyramid Convolutional Neural Network (DPCNN) for source code feature extraction and Graph Neural Network (GNN) for binary code feature extraction. These methods are simply brought from other areas. The contributions of this part are not enough. After reading the rebuttal, I want to keep my original evaluation.

Correctness: Yes

Clarity: Yes

Relation to Prior Work: Yes

Reproducibility: No

Additional Feedback:


Review 3

Summary and Contributions: This work: - sets up the problem of matching source code and binary code as a cross-modality retrieval problem; - sets up neural architectures for the two modalities and connects them in a conventional cross-modality retrieval setup; - provides an empirical comparision between the proposed approach and standard baselines.

Strengths: - This work addresses a practical and (in my view) important problem. - Empirical results seem strong. - Nice figures, e.g. Figure 1 to illustrate the problem.

Weaknesses: - The writing could be improved both on the stylistic side and on the clarity of explanations side. - The work connects standard components in a natural way, but this is not necessarily an issue. - If I understood the lines in Table 1 correctly there does not seem to be an ablation evaluating the importance of having the code literal features in the proposed approach?

Correctness: I have not spotted issues with correctness. I have not read the appendix.

Clarity: Clarity, formatting, and language could be improved. For example, the sampling method around equations (7)-(8) could be explained more thoroughly. Minor comments: - Line 137: I think "research" is usually not pluralized. - Equations (2)-(6) could be attempted to be explained in words, or a figure added. - Section 3.4: Could mention that the new hyperparameter s can be seen as a negative temperature parameter, sometimes denoted beta. - Section 3.4: Usage of the x symbol to denote standard multiplication may not be necessary.

Relation to Prior Work: Prior work is discussed rather extensively in Section 2; unfortunately I'm not familiar enough with this domain to be able to say for sure whether something was missed.

Reproducibility: Yes

Additional Feedback: - For the "source to binary" task, wwould it be even remotely feasible to try compiling the source under many combinations of target architectures and compiler options, and then look for similarities using a more primitive matching algorithm? I assume not, but a comment could perhaps be helpful. - Line 163-164: Not sure if I understand correctly, in binary code ordering may not matter, but in source code there seems to be potentially useful information in how statements follow each other. Therefore, is the stated "accordance" actually desired? - Line 169: I found the statement that only 10% of the code can be successfully parsed surprising. I would have assumed code must be parse-able if it can be compiled. Can you please explain a bit more? - Table 1: Why is the line "Random" not corresponding to a line from the 2nd block? Should the results be identical to the DPCNN+HBMP line, and the difference is due to randomness? (If so it might be helpful to add confidence intervals and/or reduce the precision of reported accuracies, perhaps to 0 decimal digits.) - Would the results get even better with a larger value of s? It might be nice to show when recall starts to decline as s is increased. - Broader impact statement: Could a potential negative outcome occur if the tools to discover known vulnerabilities in a wide range of programs are used by malicious actors?


Review 4

Summary and Contributions: The paper proposed a new model for functional-level binary source code matching. The major challenge of this task is how to learn a cross-modal embedding space. The author proposed two modality-specific encoders: a CNN-based source code encoder and a GNN-based binary code encoder. In order to deal with the string and integer literals in the data, the author proposed to use a hierarchical-LSTM to encode the string features and an integer-LSTM to encode the integer features. To better train the cross-modal embedding space, the author used the triplet loss and sampled the negatives based on the distance. The final model achieves significantly better performance than the baselines. The contribution is the introductiion of the function-level binary source code matching problem and a novel deep learning based solution to this problem.

Strengths: The paper is novel and shows that deep learning can help improve the SOTA of functional-level binary source code matching. The model design is reasonable. The ablation study in Table 1 proves the effectiveness of each design choice.

Weaknesses: This is an application paper and may lack some novelties in the modeling part. However, according to my perspective, this is a minor limitation.

Correctness: According to the best of my knowledge, the evaluation is correct.

Clarity: Yes

Relation to Prior Work: Yes, it's clearly stated in Section 2.

Reproducibility: No

Additional Feedback: The modeling design is reasonable and experiment results are very solid. However, it will be easier for others to reproduce the results if the author could provide more details about the dataset. ———— After Rebuttal ———— I choose to keep my score because the paper proposed an effective solution to a new problem. Although it lacks novelty from the modeling side, it is a good application paper.

[Author Response · NeurIPS 2020]

We thank the reviewers for their valuable feedback. We will address the comments and the concerns as follows.

**R1-Motivation.** Our motivation is to solve the binary source code matching problem, which is very important for computer security. It can be leveraged for code vulnerability analysis and malware detection. Given the binary code, the researchers want to explore its specific meaning. A general method is using a decompiler to translate the binary code into pseudo code, which requires a lot of expert experience and has low accuracy. At the same time, the final pseudo code has poor readability because of the missing variable names and the complex "goto" statements. The situation is even serious at function-level because much more information is lost, so researchers expect to find the real corresponding source code. Traditional methods use the code literals to search the source code, but with very low top1 score (40%). Our deep learning model could help capture the potential semantic features between the source/binary code and achieve 90% top 1 score. The main contribution of our paper is using deep learning models to make this problem from "cannot" to "can". Our research can benefit tens of thousands of reverse engineering researchers.

**R1&R2&R4-Novelty.** In our paper, the technique is not deep in the modeling part. However, we strongly agree with R1&R4 that this should not be taken as a limitation because getting amazing results with such a simple method proves the capability of deep learning on this task, which is exactly what we want to show. Based on our CodeCMR framework, we tried many useful novel methods, including Circle Loss, Cross-Batch Memory, Adversarial Loss, and Cross-lingual Language Model. We could add a discussion part to talk about these methods, but we still think they are not the focus of this paper because our goal is to use AI to solve valuable problems. Also, we propose the integer-LSTM for code literal modeling, and a simple but useful norm weighted sampling method which helps improve 10% top1 score.

**R1&R3-Why Treating Source Code as Text.** Treating source code as text and getting such good results has surprised many experts. In fact, it is a salient point of our paper. Before explanation, we would like to mention that "CFG" is short for "control flow graph" (a graphical representation of binary code) in our paper rather than "context free grammars", we are not sure whether this leads to the misunderstanding. The traditional approach is to parse the codes into graphical inputs. There are two non-negligible indicators in this process: time and success rate, because function-level code loses much information such as "#include" and "#define". For example, it takes about 20 seconds to parse a piece of source code into code property graph with 84.1% success rate on 10,000 pieces. These scores are unacceptable since we want to create a solid method which can be applied to our million-to-billion-level scenario rather than a toy model. Treating source code as text has no time cost and achieves 100% success rate. Surprisingly, when we compare the results of our method with parsing models using a successfully-parsed dataset, we find that our method's top1 score is only 0.5% - 1% less. Another advantage is that our method does not need much expert experience, so the AI/NLP researchers can transfer their experience on this task easily, which could help promote the development of this field.

**R1&R3-Writing.** (1) We totally agree with R1&R3 that it is more canonical to explain the equations more thoroughly. However, these sentences took up about a quarter of a page, and we had to abridge them because we want to keep other sentences for the readability and consistency of the paper. This problem could be solved easily because we will have an additional page on camera ready version. (2) For Table 1, some additional explaining sentences will be added on camera ready version too. The second section of Table 1 ignores the code literals, and the third section only uses code literals. Since we propose different models on code semantic feature and code literal feature, we should do ablation study to explore the effect of each part. And in the fourth section the RANDOM line means the combination of the best models on code semantic feature and code literal feature with random sampling method, which is DPCNN+HBMP+Integer-LSTM(Integer)+Hier-LSTM(String). By comparing the RANDOM line with DPCNN+HBMP, Integer-LSTM(Integer) and Hier-LSTM(String), we could observe that the code semantic feature is much more useful than the code literal feature, which meets our assumption. To illustrate the effect of value s, we conduct the experiments on s=8 and s=10, it turns out that s=5 is a decline point. We will add a figure to show this.

**R1&R3&R4-Dataset.** We decide to publish our dataset and introduce the details of how we build and how to use it. We first extract files from many open-source libraries including openssl, glibc, etc. Then we compile them on 32 combinations of different compilers (gcc/clang), different platforms (x86/x64/arm/arm64) and different optimizations (O0/O1/O2/O3). About 80,000 files are successfully compiled, then we split them into about 1,700,000 functions. We randomly sample 50,000 functions and split them into trainset, validset, and testset (3:1:1). The functions are saved in a .pkl file with 33 columns, in which the first column contains source code, and 2nd-33rd columns contain binary code.

**R3-Details.** (1) For "source to binary" task, it is feasible to compile and compute similarity scores. Our method is more powerful. Firstly, we could save much time from compiling the code under many combinations, because our method could achieve amazing results on different combinations (more details could be found in supplementary material Figure 1). Secondly, sometimes the function-level code could not be successfully parsed because of information loss, while our method could always work. (2) The source code order is not useful on high-level. For example, "if A else B" equals to "if B else A". We have tried positional encoding and LSTM to capture the order feature, but the score declines. (3) We think the answer of the boarder impact question is yes, and we will add it on camera ready version. However, we believe that our research has more positive outcomes because the majority of researchers are protectors rather than attackers.

[Meta-Review · NeurIPS 2020]

Overall, the reviewers have mixed opinions about this work. Although everyone agreed that this is an application paper, the reviewers felt that there are a series of issues that still need to be addressed. Specifically, in a camera-ready the authors are kindly asked to include "Circle Loss, Cross-Batch Memory, Adversarial Loss, and Cross-lingual Language Model" (as promised in the author feedback) to alleviate some of the reviewer concerns. Nevertheless, this paper addresses a problem in an interesting and important application domain and as such exploring how known neural architectures used in other domains can be adapted and used will be useful for the community, hence I believe this paper should be accepted.